# Understanding Osteochondritis Dissecans: A Narrative Review of the Disease Commonly Affecting Children and Adolescents

**DOI:** 10.3390/children11040498

**Published:** 2024-04-22

**Authors:** Wojciech Konarski, Tomasz Poboży, Klaudia Konarska, Michał Derczyński, Ireneusz Kotela

**Affiliations:** 1Department of Orthopaedic Surgery, Ciechanów Hospital, 06-400 Ciechanów, Poland; tomasz.pobozy@onet.pl (T.P.); michalderczynski@gmail.com (M.D.); 2Medical Rehabilitation Center, Sobieskiego 47D, 05-120 Legionowo, Poland; klodiii87@gmail.com; 3Department of Orthopedic Surgery and Traumatology, Central Research Hospital of Ministry of Interior, Wołoska 137, 02-507 Warsaw, Poland; ikotela@op.pl

**Keywords:** osteochondritis dissecans, OCD, knee, elbow, ankle

## Abstract

Background: Osteochondritis dissecans (OCD) is a joint disorder predominantly affecting the knee, elbow, and ankle of children and adolescents. This comprehensive review delves into the epidemiology, etiology, clinical manifestations, diagnostic approaches, and treatment of OCD. Results: The most common cause of OCD is repetitive microtrauma, typically associated with sports activities, alongside other significant factors such as genetic predisposition, ischemia, and obesity. In early stages or when lesions are small, OCD often presents as non-specific, vaguely localized pain during physical activity. As the condition progresses, patients may experience an escalation in symptoms, including increased stiffness and occasional swelling, either during or following activity. These symptom patterns are crucial for early recognition and timely intervention. Diagnosis in most cases is based on radiographic imaging and magnetic resonance imaging. Nonsurgical treatment of OCD in young patients with open growth plates and mild symptoms involves activity restriction, immobilization methods, and muscle strengthening exercises, with a return to sports only after symptoms are fully resolved and at least six months have passed. Surgical treatment of OCD includes subchondral drilling in mild cases. Unstable lesions involve methods like restoring the joint surface, stabilizing fractures, and enhancing blood flow, using techniques such as screws, anchors, and pins, along with the removal of fibrous tissue and creation of vascular channels. The specifics of OCD treatment largely depend on the affected site. Conclusions: This synthesis of current research and clinical practices provides a nuanced understanding of OCD, guiding future research directions and enhancing therapeutic strategies.

## 1. Introduction

Osteochondritis dissecans (OCD) is a joint disorder characterized by the acquired impact on both the articular surface and subchondral bone. In 1888, Koönig introduced the term OCD and hypothesized an inflammatory origin for the disease. While contemporary perspectives now associate OCD with mechanical stress, the exact etiology remains elusive [1,2]. The Research on Osteochondritis Dissecans of the Knee (ROCK) group has introduced a novel definition, characterizing OCD “a focal idiopathic alteration of subchondral bone and/or it’s precursor with risk for in-stability and disruption of adjacent articular cartilage that may result in premature osteoarthritis” [3]. Although the disease can occur at any age, it is evidently more prevalent among children and adolescents [4]. In young individuals, OCD frequently seems to be associated with disturbances in bone growth and development, potentially resulting from repetitive microtrauma or vascular irregularities. There may also be factors related to problems in endochondral ossification in juvenile OCD. In adults, OCD is commonly believed to stem from juvenile OCD that has not been resolved, or it can emerge spontaneously due to ongoing stress and strain, which may cause fractures in the subchondral bone. Additionally, adults are more likely to experience significant effects from degenerative changes and repeated usage [5].

Distinguishing between juvenile and adult forms of OCD is crucial for prognostic considerations. In those with an open growth plate, OCD is classified as the juvenile form, generally associated with a more favorable overall prognosis. Conversely, the adult variant of OCD is diagnosed in individuals with a closed growth plate, indicating a less optimistic outlook. The adult form often necessitates surgical intervention for successful healing, as spontaneous recovery is rare [2,6,7].

Multifaceted characterization of OCD underscores the dynamic interplay of factors contributing to its pathogenesis, clinical presentation, and outcomes, driving continued research efforts toward a more comprehensive understanding and effective management of this intriguing joint disorder. Therefore, aim of this narrative review is synthesis of the current knowledge evolving perspectives, and current research on OCD. In essence, this paper contributes to the ongoing discourse on OCD by offering a comprehensive overview that bridges historical perspectives with contemporary research. It aims to inform clinical practice, stimulate further research, and ultimately improve the care and prognosis for individuals affected by this disorder.

## 2. Methods

To ensure a thorough narrative review of OCD, a deep search of the relevant literature was conducted. The primary database utilized for this search was PubMed. The search strategy involved the use of relevant keywords and phrases specifically tailored to OCD. These keywords were carefully selected to encompass various aspects of the condition, including its etiology, pathology, clinical presentations, diagnostic approaches, and management strategies. We searched keywords in titles and abstracts. An example phrase for epidemiology was (epidemiology [Title/Abstract]) AND (osteochondritis dissecans [Title/Abstract]). In addition to the PubMed search, the reference lists of the selected articles were thoroughly examined.

To be included in this review, studies were required to be peer-reviewed articles written in English, with available full-texts, presenting findings that are relevant and recent to the field. We aimed to incorporate the most recent literature; however, a specific date range was not applied, as some treatment methods have not evolved significantly over time. For the treatment section, clinical studies were included, but case reports were not. In other sections, we allowed citation reviews. Studies were excluded if they did not directly focus on the identified topics of this review or were in the form of preliminary reports, conference abstracts, or unpublished manuscripts, ensuring the quality and completeness of the data included.

Relevant data from the included studies were extracted and organized into thematic categories. This thematic analysis facilitates a narrative synthesis, allowing for a discussion that not only presents the findings but also interprets and contextualizes them within the broader literature.

## 3. Epidemiology

The results of the ROCK group’s analysis of a multicenter prospective cohort indicated that the most commonly found OCD lesions are located in the knee and elbow, specifically in medial femoral condyle (66.2%), followed by the lateral femoral condyle (18.1%), trochlea (9.5%), patella (6.0%), and tibial plateau (0.2%) [8]. Ankle OCD is less common with lesions located in the posteromedial (71.8%), anterolateral (22.4%), or central (3.5%) regions of talus [9].

According to the literature, the prevalence of knee OCD ranging from 2.3 to 31.6 cases per 1000,000 population [10,11]. The highest prevalence is observed in juveniles and reaches 11.2 per 100,000 in the 12-to-16-year-old age group [11]. The multivariable logistic regression analysis indicated a 3.3-fold increased risk of knee OCD in patients aged 12 to 19 years compared to those aged 6 to 11 years (*p* < 0.001). Moreover, male patients exhibited a 3.8-times-higher risk of knee OCD than females (*p* < 0.001) [11]. Most cases of knee OCD involved multisport athletes, with a particularly high prevalence among male basketball players and female soccer players [8]. Concerning elbow OCD, data from the United States [12] showed an incidence of capitellar OCD at 6.0 per 100,000 in the general population, 9.5 per 100,000 for male patients, and 2.6 per 100,000 for female patients. The incidence was particularly elevated in younger patients. Among the affected individuals, 89% were engaged in sports, with 90% being overhead athletes and 58% identified as throwing athletes [12]. The frequency of ankle OCD in individuals between 6 and 19 years old was overall 4.6 cases per 100,000. Specifically, the rates were 3.2 per 100,000 in male patients and 6.0 per 100,000 in female patients. Additionally, a multivariate logistic regression analysis indicated that the risk of ankle OCD was 6.9 times higher in patients aged 12 to 19 years compared to those aged 6 to 11 years [9]. OCD of the shoulder and hip are relatively uncommon conditions, with shoulder OCD being more prevalent. It accounts for 1.6% of OCD cases found in children [13].

Yellin et al. [14] examined the medical records of 80 patients aged ≤18 years diagnosed with unilateral knee OCD. Contralateral knee imaging of the asymptomatic knee, performed within 1 year of the initial presentation (a routine practice for several physicians in the group), was necessary to identify asymptomatic contralateral knee lesions. The study revealed that 15% of patients exhibited bilateral disease, with no significant distinctions in age, sex, physical status, or lesion characteristics between those with unilateral versus bilateral conditions [14].

## 4. Etiology

The precise etiology of OCD remains unclear. OCD typically begins as an injury to the subchondral bone, which then progresses through various stages. These stages include bone resorption, followed by the bone collapsing, and ultimately the formation of a sequestrum. This process can eventually result in the articular cartilage separating and the detachment of a subchondral bone fragment, finally leading to the formation of a loose body [10].

It is generally believed that OCD develops due to mild, repetitive microtrauma, often associated with sporting activities. This repetitive stress can lead to vascular issues and stimulate a subchondral response, which might disrupt the healing of the bony trabeculae, ultimately hindering the bone’s capacity to recover [2,15,16,17]. In the context of sports, capitellar OCD is mainly observed in the dominant arms of athletes involved in throwing sports and gymnastics, where they experience valgus or axial stress [18]. Juvenile OCD of the femoral trochlea is frequently seen in athletes who play basketball, football, and soccer, as these sports expose the patellofemoral joint to significant forces [17,19,20]. Some studies show connection between discoid meniscus and mechanical axis malalignment lateral femoral condyle and the presence of OCD [1,21].

Another causal factor of OCD is local ischemia. Several studies have suggested that inadequate blood supply and resulting ischemia might be a contributing cause of OCD. Differences in vascular patterns have been observed at sites prone to OCD. A combination of such joint structures, along with repeated trauma could lead to ischemic events, potentially leading to the development of OCD [22,23].

There is an established link between childhood obesity and the occurrence of OCD in the knees, ankles, and elbows of children. Research by Kessler et al. indicates that extremely obese patients face an 86% increased risk of developing any type of OCD compared to individuals of normal weight [24].

Some researchers suggest that familial inheritance might play a role in the development of OCD, noting higher occurrence rates in family cases and among identical twins [25,26]. Yellin et al. [25] proposed specific genetic markers that could be crucial in understanding OCD’s pathophysiology. They discovered a set of genetic variations called single-nucleotide polymorphisms (SNPs) on chromosome 13, along with a prominent signal on chromosome 7, which are likely important in the coordinated gene expression associated with OCD. Additionally, they identified an SNP on chromosome 12, specifically rs1464500, located in the *SOX5* gene. This gene is thought to play a significant role in the development of chondrocytes, the cells vital for forming cartilage [27].

## 5. Manifestation Picture and Diagnosis

### 5.1. Manifestation

The symptoms of OCD vary significantly based on the location and stage of the disease. Stable lesions typically lead to general symptoms, including swelling, activity- or palpation-related pain, vague crepitus, restricted movement range, and joint effusion. In contrast, unstable lesions or loose bodies can cause symptoms like catching, clicking, or locking of the joint [17,28,29].

In early stages or when small, OCD lesions in the knee often manifest as non-specific, vaguely localized pain during physical activity [30]. As the condition advances, the patient may experience a gradual increase in stiffness and occasional swelling, either during or following activity. In cases of advanced or larger lesions, there might be symptoms of catching or locking, particularly if there is a loose foreign body in the joint. However, a loss of range of motion is not typically observed in these cases [2,31].

Elbow OCD has vague symptoms which often delays diagnosis. The usual profile for a patient with this condition is a young male athlete who first shows symptoms of initially mild pain but which improves with rest, followed by tenderness, and swelling on the outer side of the elbow. In the later stages of the condition, the patient may experience a loss of extension in the elbow and intermittent episodes of catching and locking [32,33].

Patients with talus OCD commonly have a history of an ankle inversion injury. The condition may remain symptom-free for an extended period. Symptomatic cases typically involve intermittent pain during weight-bearing activities such as running. If the osteochondral fragment detaches, the symptoms become more severe and include intense pain (often described as an “articular crisis”), swelling in the joint, instability while walking, and potentially locking of the joint [28,34].

### 5.2. Physical Examination

When dealing with injuries to the medial condyle of the femur, the Wilson test can be used. This test typically causes pain when extending the knee from a 90° to 30° angle while simultaneously rotating the tibia internally. However, this pain tends to decrease when the tibia is rotated externally [35,36]. In contrast, for injuries to the capitellar joint, the radiocapitellar compression test may be applied. This test induces pain during active pronation and supination movements of the forearm while the elbow joint is fully extended [37]. In their case series, Conrad and Stanitski evaluated the validity of Wilson’s test in a group comprising 17 juvenile and 15 adolescent patients with knee OCD. Out of the 32 patients, 24 (75%) who had radiographically confirmed OCD at their first visit showed negative results on Wilson’s test. The other eight patients, who initially tested positive, showed a shift to negative results as their lesions resolved. The authors concluded that Wilson’s test has minimal value for clinical diagnosis. However, when positive, it can serve as a useful tool for monitoring the progress of the condition during treatment [35].

In cases of elbow OCD, there may be swelling of the posterolateral elbow plica and tenderness upon palpation at the anterolateral aspect of the elbow. This can occur with or without impingement and may be accompanied by a loss of terminal extension. In more advanced stages, provocative tests targeting the lateral compartment of the elbow often induce pain. However, forearm pronation and supination movements are typically not restricted. The ‘grip and grind’ test, a specific provocative test for the radiocapitellar joint, may yield positive results, indicated by crepitus or a clicking sensation. The presence of elbow locking may suggest the formation of a loose body. Currently, there is no available evidence regarding the sensitivity and specificity of these tests [33].

When examining ankle OCD, it is crucial to check for any localized tenderness. To palpate the talar dome effectively, the ankle should be placed in extreme dorsiflexion. Additionally, assessing the range of motion and any potential associated ligamentous laxity is important. This assessment should include a comparison with the opposite (contralateral) side [28].

### 5.3. Imaging

#### 5.3.1. Radiography

Radiography is the primary method for diagnosing and tracking the progress of OCD treatment. It is also advisable to conduct bilateral and standing alignment radiographs since bilateral OCD of the knee may occurs in 14–30% cases [17,38,39]. Early stages of OCD may show changes in the contour and radiolucency near the joint surface on radiographs (Figure 1A). More progressed stages often reveal a distinct, sometimes ossified fragment (referred to as the progeny), separated from the main bone (the parent) by a crescent-shaped radiolucent line, which might ossify as the healing progresses [17].

In knee OCD, an antero-posterior view, a lateral view, and a notch view of the knee should be obtained [17]. When there is a suspicion of an OCD lesion in the patella or trochlea, obtaining a skyline view through radiography is essential. The usual radiographic presentation of such a lesion includes a well-defined area of subchondral bone, which is distinguished by a combined sclerotic and radiolucent border surrounding the fragment. While radiographs are valuable for diagnosing these lesions and tracking their healing process, they fall short in evaluating the fragment’s viability and the condition of the interface between the subchondral bone and cartilage. Additionally, radiographs are not effective in predicting the stability of the fragment [2]. It is important to distinguish knee OCD from the normal variation in irregular contours on the posterior femoral condyle, which is commonly observed in patients with open growth plates (physes), particularly between the ages of 6 and 10. Unlike OCD, this normal irregularity does not exhibit a surrounding sclerotic rim and tends to become less pronounced over time [40].

In elbow radiography, specific signs indicative of OCD of the capitellum include a flattened capitellum, a distinct defect on the articular surface, and the presence of loose bodies. However, standard elbow radiographs are often not sufficiently sensitive to detect capitellar OCD. In fact, nearly half of the patients with this condition have normal-appearing radiographs. For a more effective assessment, recommended radiographic views of the elbow include anteroposterior views in both full extension and at 45° flexion, lateral views, and external oblique views. Notably, an anteroposterior view with the elbow at a 45° flexion angle can be more revealing of the lesion compared to an anteroposterior view in full extension [17,33,41,42].

In the case of ankle OCD, an antero-posterior projection frequently reveals a subchondral halo. This sign, highlighting the osteochondral fragment, also aids in distinguishing OCD from other conditions such as an intraosseous mucoid cyst or a dystrophic lesion. Additionally, a radiograph taken with the ankle positioned at 15° of internal rotation is especially beneficial for examining the supero-lateral corner of the talus. This specific positioning ensures that the area is clearly visible, free from any overlapping by the fibula [28].

While radiography is effective in pinpointing the location and size of the lesion, it is less accurate in assessing the stability of the lesion and in detecting smaller, more subtle lesions [17].

#### 5.3.2. Magnetic Resonance Imaging

Magnetic resonance imaging (MRI) is the preferred diagnostic tool for OCD. It offers precise measurement of the lesion size, detection of bone edema, and identification of any intra-articular loose bodies [43].

In knee OCD, the progeny typically appears hypointense on T1 MRI images and shows heterogeneous signals on T2 images (Figure 2). MRI is more effective in evaluating the volume of the lesion and may reveal an osteochondral fragment that extends beyond the normal contour of the epiphysis, a defect at the original site of the fragment, or loose fragments within the joint cavity. The “Oreo cookie sign” in MRI imagery is characterized by a curvilinear hyperintense T2 signal at the interface of the progeny and the parent bone (resembling the cream of an Oreo cookie), flanked by two layers of hypointense signals (similar to the cookie wafers). The presence of focal cysts at the interface between the progeny and the parent bone indicates a more chronic condition [40].

In elbow OCD, MRI is effective for diagnosing the disease in its early stages. However, it is less sensitive than computed tomography (CT) in detecting loose bodies in more advanced cases of OCD [33].

In OCD of the talus, the osteochondral fragments display a hypointense signal on T1-weighted MRI images. On T2-weighted images, these fragments present with varying signal intensities, but they are consistently marked by a hyperintense line at their base. This line is indicative of fragment detachment [28,44].

#### 5.3.3. Computed Tomography

CT scans are valuable for assessing the size and location of lesions, detecting loose bodies, and especially for monitoring bone healing post. This imaging technique is frequently utilized for examining OCD in the capitellum and navicular bone [17]. Nevertheless, conventional CT is less effective in evaluating articular cartilage and other non-calcified components of a joint [45]. These assessments can instead be achieved through CT-arthrography [17].

#### 5.3.4. Other Techniques

In the past, scintigraphy was utilized for diagnosing OCD and to evaluate healing by measuring perfusion. Although it was highly sensitive, scintigraphy has fallen out of favor due to its lack of specificity, the widespread availability of MRI, and concerns related to exposure to radioisotopes [17,45,46,47].

The use of ultrasound scanning for the elbow has been documented as a method to detect capitellar OCD, and it can serve as a useful screening tool outside of hospital settings. The effectiveness of the ultrasound scan greatly relies on the experience and skill of the examinator [33,48,49].

### 5.4. Differential Diagnosis

The differential diagnosis for OCD varies between pediatric patients and adults, with specific details outlined in Table 1.

### 5.5. Classifications

Table 2 displays the classifications of OCD severity, which are based on findings from radiography and MRI, for the most commonly affected OCD sites.

## 6. Treatment

### 6.1. Conservative Treatment

The recommended approach for treating OCD without surgery focuses on younger patients with open growth plates and mild symptoms [55]. This strategy primarily involves limiting physical activities and sports participation. The athlete is advised not to resume playing for a minimum of six months and should only consider returning to play once all symptoms have fully resolved [55]. Treatment methods include using casts, braces, or splints for immobilization, restricting weight-bearing activities, performing exercises to strengthen muscles, and physiotherapy such as extracorporeal shock wave therapy (ESWT) [17,28,43,56]. This nonsurgical treatment is typically applied for a period ranging from 3 to 6 months [17,28,43]. In the systematic review by Andriolo et al. [57], the authors summarized five distinct treatment approaches of conservative treatment: restricting physical activity, physiokinesitherapy with muscle-strengthening exercises, physical instrumental therapies (including iontophoresis and ESWT), limiting weight-bearing (either partial with crutches or total with a wheelchair), and immobilization (using a cast or brace). The review revealed an overall healing rate of 61.4% (487 out of 793 patients), with success rates varying from 10.4% to 95.8%, after excluding case reports and studies involving fewer than five patients. It should be noted that higher age is associated with a poorer prognosis, suggesting that conservative treatment methods are generally more advisable for younger patients with immature skeletons [57].

Kocher et al. suggested a three-phase approach for non-surgical management of knee OCD. The first phase involves immobilization and partial weight-bearing using crutches for 4–6 weeks. Following this, phase two commences with weight-bearing without immobilization after a radiographic check, along with a rehabilitation program focusing on muscle strengthening and full-range-of-motion recovery for an additional 6–12 weeks. If healing is evident both radiographically and clinically three to four months post-diagnosis, phase three allows a gradual return to sports with a follow-up MRI [2,10].

Regarding pharmacological treatment of OCD, it appears that there is limited direct research focusing on specific medications for treating the condition itself. Some authors suggest using nonsteroidal anti-inflammatory drugs as a conservative treatment method for managing OCD symptoms [58,59].

### 6.2. Surgical Treatment

If pain persists or worsens after six months of conservative treatment, or if there are signs of lesion instability on an MRI or a lack of radiographic healing, surgical intervention is recommended [2]. Surgical treatment of OCD depends on the location of the affected area.

#### 6.2.1. Knee OCD

In milder cases, a technique known as subchondral drilling, either through the joint (transarticular) or behind the joint (retroarticular), is frequently used. This approach ensures the flow of mesenchymal cells and growth factors, which results in the formation of new blood vessels and the transformation of the cartilage defect into health bone [43,60]. Numerous reports confirm the positive outcomes associated with subchondral drilling [61,62,63,64,65,66,67].

Unal et al. conducted a retrospective study on 41 knees affected by OCD that were treated using anterograde chondral drilling. Six months after the operation, 78% knees experienced a full alleviation of symptoms. Radiographic healing was noted in 66% of the cases. Patients who were symptom-free at six months resumed sports activities at a similar level as before, with a mean return time of 7.9 months. Conversely, those who still had symptoms at six months took an average of 16.5 months to return to their sports activities [68].

In research conducted by Adachi et al., 20 OCD lesions in knees of 12 young patients with growing skeletons were treated using retroarticular drilling. There was a notable post-surgical improvement in the average Lysholm score, increasing from 72.3 to 95.8. Except for one, all lesions healed following the retroarticular drilling procedure. The average time to observe healing was 4.4 months on standard radiography and 7.6 months when assessed via MRI [61].

In their systematic review, Gunton et al. found that both retroarticular and transarticular drilling techniques, when applied to stable lesions, yielded similar outcomes in terms of short-term patient-reported results and radiographic evidence of healing [66]. The latest research by the ROCK Group found that transarticular drilling of knee had shorter operation and fluoroscopy durations and better healing indicators at 6 and 12 months compared to retroarticular drilling. However, there were no significant differences in healing parameters at 24 months or in patient-reported outcomes at any stage of follow-up between the two methods [69]. In their retrospective analysis of 131 patients, Baghdadi et al. propose that arthroscopic drilling for stable, intact osteochondritis dissecans (OCD) lesions in the pediatric knee is a safe technique, offering dependable results, including resumption of activities of daily living (ADLs) with a minimal risk of complications. The majority of patients were able to return to their pre-surgery level of daily activities and achieve full knee mobility within three months post-operation.

In their retrospective analysis of 131 patients, Baghdadi et al. [70] proposed that arthroscopic drilling for stable knee OCD in pediatric patients is a safe technique, offering dependable results, including resumption of activities of daily living with a minimal risk of complications. The majority of patients (95.7%) showed healing on radiographs at the three-month postoperative visit. The management of unstable lesions encompasses restoring the joint surface, stabilizing the fracture, and improving blood flow. There are a variety of methods to secure unstable OCD lesions, such as the use of screws, anchors, arrows, and pins. These can be inserted either through arthroscopic procedures or in open surgeries. An essential part of this process is the meticulous removal of fibrous tissue and the creation of vascular channels through drilling. This step aims to boost blood supply to the affected area and improve healing [2,43].

Husen et al. conducted a multicenter retrospective cohort study that included 25 skeletally immature patients and 56 patients with closed growth plates, all treated with internal fixation for unstable knee OCD. After an average follow-up period of 11.3 ± 4 years, lesions had healed in 58 (71.6%) patients, while 23 (28.4%) patients experienced non-healing lesions. The study found no significant difference in the risk of healing failure based on the status of physeal maturation [71].

For patients with the most severe forms of the disease, where fixation is not an option, recommended treatments include methods like microfracturing, autologous chondrocyte implantation, bone marrow stimulation, the use of fresh osteochondral allografts, or autologous chondrocyte transplantation [2,17,43]. Gudas et al. compared the outcomes of knee OCD patients treated with either osteochondral autologous transplantation or microfracture procedures. The study involved 60 athletes, averaging 24.3 years old, with symptomatic knee cartilage lesions, who were randomly assigned to either treatment. After 37.1 months, both groups showed significant clinical improvement (*p* < 0.05). Functional and objective assessments, based on the modified Hospital for Special Surgery and International Cartilage Repair Society scores, revealed that 96% had excellent or good results with osteochondral autologous transplantation, in contrast to 52% for the microfracture procedure (*p* < 0.001). At 12, 24, and 36 months post-surgery, the osteochondral autologous transplantation group consistently showed significantly better results [72].

Ogura et al. [73] reported on the treatment outcomes for six adolescent patients with OCD of the knee, utilizing autologous bone pegs for chondral fragment fixation. Five of these patients were able to resume sports activities without any limitations, averaging a return at seven months post-surgery (ranging between six to eight months). At their most recent follow-up, these five individuals demonstrated full knee mobility and exhibited no signs of joint effusion.

Komnos et al. [74] described the outcomes of treating 40 juvenile patients with knee OCD using arthroscopic retrograde drilling and internal fixation with bioabsorbable pins. MRI results confirmed lesion healing in 36 out of the 40 patients (90%). Specifically, the healing rates were 95% (20 out of 21 patients) for stage II lesions and 84% (16 out of 19 patients) for stage III lesions.

Baldassarri et al. [75] used a one-step bone marrow-derived cell transplantation technique on 18 patients with knee OCD. The IKDC and KOOS clinical scores exhibited a progressive increase. The Tegner Score at the final follow-up (5.3 ± 2.7) was significantly lower compared to the pre-injury level (6.5 ± 2.1). However, larger sample sizes and more extensive follow-up evaluations are needed to confirm these results.

Using autologous chondrocyte transplantation to treat OCD lesions in the knee results in integrated repair tissue and achieves successful clinical outcomes even in 90% of patients [76]. In the randomized clinical trial by Gudas et al. [77] with a follow-up period averaging 4.2 years, children under 18 years of age demonstrated significantly better outcomes with mosaic-type osteochondral autograft transplantation (OAT) compared to microfracture (MF) for treating knee OCD. Nevertheless, this research indicates that both MF and OAT yield promising clinical outcomes for children under 18 years of age.

#### 6.2.2. Elbow OCD

For patients who continue to experience symptoms despite taking rest and modifying their activities, or for those with unstable lesions, surgery is recommended. A variety of surgical approaches for treating capitellar OCD lesions have been documented. These approaches range from arthroscopic or open in situ fixation, to debridement with or without drilling or microfracture, and even osteochondral autograft transfer from the knee [78].

Braig et al. analyzed the outcomes of patients with elbow OCD who underwent either surgical or conservative treatment. Out of 50 elbows with a median follow-up of 10.3 years, 14% were treated nonoperatively, 32% received delayed surgery after at least 6 months of unsuccessful nonoperative treatment, and 54% had early surgical intervention. The study found that compared to nonoperative care, surgical treatment led to better Mayo Elbow Performance Index pain scores (40.1 vs. 33; *p* = 0.04), fewer mechanical symptoms (9% vs. 50%; *p* < 0.01), and improved elbow flexion (141° vs. 131°; *p* = 0.01) in the long term [79].

In cases of grade 2 lesions, where there is a separation between the OCD fragment and the adjacent bone, fragment fixation can be considered. For larger lesions that contain a thicker bone segment within the OCD fragment, methods such as compression, Herbert’s screws, or bioabsorbable screws may be employed for securing the fragment [78,80].

Kuwahata et al. [81] documented the treatment of eight elbows afflicted by OCD of the capitellum, utilizing cancellous bone grafts and internal fixation with a Herbert screw. At an average follow-up of 32 months post-surgery, all patients reported being free from pain. Smaller lesions that are not suitable for screw fixation can be managed using the pull-out wire technique. Takeda et al. [82] documented the treatment of 11 male baseball players using the pullout wiring technique and bone grafting. All individuals experienced pain relief. The wires were extracted on average 17 weeks post-operation. Follow-up radiographs indicated healing and the absence of degenerative changes in the radiocapitellar joint.

As arthroscopic techniques have advanced, debridement has emerged as the cornerstone of surgical intervention for capitellar OCD that do not respond to activity modification or in cases where the OCD lesions are unstable. In the retrospective analysis by Brownlow et al. [83] of 29 patients who underwent arthroscopic debridement for capitellar OCD. The patients’ average age at the time of surgery was 22 years. At an average follow-up of six years, all individuals could carry out daily activities, and 28 out of 29 patients described their outcomes as good to excellent. Other authors have also confirmed positive outcomes from debridement in the treatment of elbow OCD [84,85].

For patients presenting with large defects, the osteochondral autograft transfer technique, employing either single- or multiple-bone and cartilage plugs (known as mosaicplasty), may be advantageous. Iwasaki et al. [86] reported on a cohort of 19 teenage male competitive athletes suffering from advanced capitellar OCD. Following the procedure, 18 of these patients reported no elbow pain, while one reported occasional mild pain.

Some studies have documented the treatment of elbow OCD using regenerative approaches. Guerra et al. [87] detailed the treatment of three juvenile patients with elbow OCD using bone marrow-derived cells. All three patients demonstrated clinical progress, including a minor improvement in the range of motion by the time of the last follow-up. Farr et al. [88] reported on five adolescents with grade 3–4 elbow OCD who underwent treatment involving debridement, transplantation of cancellous bone from the iliac crest, and the application of a hyaluronic acid-based scaffold. All patients achieved good to excellent clinical outcomes, with complete resolution observed in only 2 of the 5 cases. There was a slight improvement in elbow motion post-surgery. No complications were reported.

#### 6.2.3. Ankle OCD

Patients who are skeletally immature and those with early-stage lesions of the talus typically have a favorable response to non-surgical treatment [89,90]. Choi et al. [91] demonstrated that arthroscopic microfracture treatment for talus OCD yields positive functional results. In their study, 165 osteochondral lesions, averaging 73 mm^2^ in size, treated with microfracture saw notable enhancements in functional scores over a period of 6.7 years. Of the 165 ankles treated, 22 (13.3%) required subsequent arthroscopic surgeries to assess the condition of the repaired cartilage. Subchondral drilling showed results comparable to those of microfracture treatment. [92,93].

For larger lesions, advanced cartilage restoration techniques, such as autologous chondrocyte implantation, are accessible. A systematic review conducted by Erickson et al. [94] found no significant difference between open and arthroscopic autologous chondrocyte implantation techniques.

## 7. Clinical Applications

The management of OCD involve a multidisciplinary team including a radiologist, orthopedic surgeon, physical therapist, nurse practitioner, and primary caregiver. The significance of promptly diagnosing and properly managing OCD is crucial. Various classification systems are used to evaluate the lesions, focusing primarily on the involvement and mobility of the overlying cartilage. Delayed or insufficient treatment can result in joint deterioration, early-onset osteoarthritis, and lasting functional limitations. Treatment decisions are based on the patient’s age, timing of diagnosis, symptom severity, and lesion stability. For stable lesions in younger patients, conservative treatment with immobilization and protected weight-bearing is preferred, the duration of which depends on the affected joint. If conservative treatment fails in stable lesions, drilling techniques (either retroarticular or transarticular) may be used, showing high healing and symptom improvement rates, with transarticular drilling typically having higher success rates. Unstable or displaced lesions require surgical intervention, usually through arthroscopic methods. Generally, stable lesions have better outcomes compared to unstable ones [95,96,97].

## 8. Conclusions

This article offers a thorough exploration of the knowledge and strategies involved in managing OCD, emphasizing its complex nature and the relationship between genetic predispositions and environmental elements like physical activity. Early detection and appropriate treatment are critical in safeguarding adolescents from the fragmentation of OCD lesions and preventing permanent cartilage damage. This comprehensive understanding allows for more effective and tailored approaches to treatment, potentially leading to better outcomes for patients. Moreover, it opens avenues for further research and advancements in OCD management techniques.

There is a critical need for increased awareness among healthcare professionals and the public regarding OCD, its risk factors, early signs, and treatment options. Such efforts are essential for earlier diagnosis and improved outcomes. This manuscript offers valuable perspectives for clinicians, researchers, and patients, contributing significantly to the ongoing effort to enhance understanding and management of OCD.

## Figures and Tables

**Figure 1 children-11-00498-f001:**
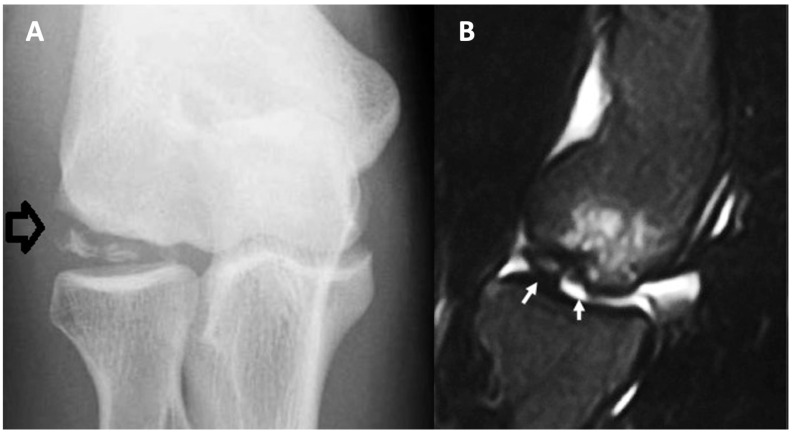
Radiographic (**A**) and computed tomography (**B**) pictures of humeral OCD. Arrows indicate OCD in the humeral head.

**Figure 2 children-11-00498-f002:**
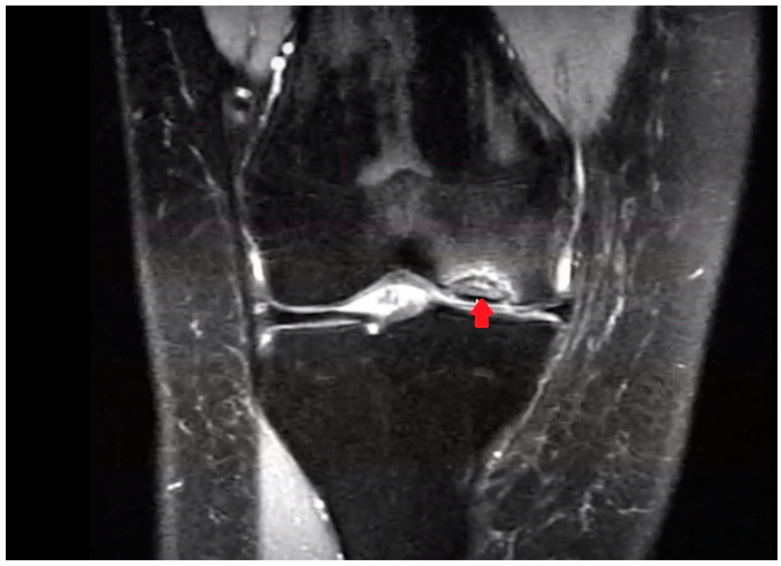
OCD of the medial femoral condyle as seen on an MRI (coronal view, T2-weighted). The arrow indicates OCD of the medial condyle of the femur.

**Table 1 children-11-00498-t001:** Differential diagnosis of OCD.

Pediatric Population	Adult Population
Patellofemoral syndrome Patellar tendonitis Osgood–Schlatter disease Sinding-Larsen–Johannson syndrome Fat pad impingement Symptomatic discoid meniscus Symptomatic synovial plica	Patellofemoral pain Knee osteoarthritis Chondromalacia Avascular necrosis Patellar tendonitis Meniscal tear Fat pad impingement Symptomatic synovial plica

**Table 2 children-11-00498-t002:** OCD classification systems.

Stage/Type	Radiography	MRI
Knee OCD [17,50,51]		
I	No changes	Thickening of the articular cartilage and low signal changes observed, yet no fractures identified
II	Sclerosis	The articular cartilage is compromised, accompanied by a low-signal rim behind the fragment, indicative of a fibrous attachment
III	Partial loosening	The articular cartilage is compromised, accompanied by a high-signal rim behind the fragment, which suggests the presence of synovial fluid between the fragment and the underlying subchondral bone
IV	Full detachment or loose body	Loose body
Elbow OCD [17,50,52]		
I	A clear, cyst-like shadow observed on either the lateral or central part of the capitellum	Similar to knee OCD
II	A distinct separation or fissure is visible between the lesion and the adjacent subchondral bone	
III	Loose bodies	
IV	-	
Talus OCD [17,53,54]		
I	Similar to knee OCD	Damage confined solely to the articular cartilage
II		IIA—Injury to the cartilage accompanied by an underlying fracture and associated bone edema IIB—Stage IIA lesion without any associated bone edema
III		Detached nondisplaced fragment
IV		A fragment that is both detached and displaced
V	-	Creation of subchondral cyst

## Data Availability

The original contributions presented in the study are included in the article, further inquiries can be directed to the corresponding author.

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
