# Peer review of "Understanding Osteochondritis Dissecans: A Narrative Review of the Disease Commonly Affecting Children and Adolescents"

_children, 2024, doi:10.3390/children11040498_

Round 1
Reviewer 1 Report
Comments and Suggestions for Authors
General
Nice overview of OCDs. The disease is rare in the general population and therefore overviews like this are helpful to highlight what is known.
Introduction
ROCK definition has changed slightly.
“a focal idiopathic alteration of subchondral bone and/or it’s precursor with risk for instability and disruption of adjacent articular cartilage that may result in premature osteoarthritis”
Epidemiology
· The epidemiology in this rare disease is difficult to be specific and certain about. The increased risk in blacks is seen in a single study and likely biased due to the population surveyed. I would therefore qualify this statement.
· This is also true with the statement about basketball as the primary sport.
· The picture and MRI of the hip ‘OCD’ is interesting. The possibility that this is a true OCD is low and the chondral shearing does not appear to be the type of pathology seen. Given that the hip is perhaps the most rare location for OCDs I would suggest this be eliminated from the manuscript to reduce confusion.
Clinical Picture
· This section is well written
· Again, the example of a navicular OCD is confusing. Interesting but confusing and may not be an actual OCD, in my opinion. I would consider eliminating this example.
· Classification systems – Classification systems are still evolving and are somewhat confusing when compared to one another. Using a classification system from 1959 is not, in my opinion, helpful except for historical purposes.
Treatment
· Non-operative treatment for stable lesions in young patients is often successful. The limited discussion of this should be expanded.
· The surgical treatment of OCDs is detailed. It is difficult to understand and perhaps could be combined into a table.
Conclusion
· Agreed
Comments on the Quality of English LanguageSome minor proof reading needed
Author Response
General
Nice overview of OCDs. The disease is rare in the general population and therefore overviews like this are helpful to highlight what is known.
Authors: Thank you for this review.
Introduction
ROCK definition has changed slightly.
“a focal idiopathic alteration of subchondral bone and/or it’s precursor with risk for instability and disruption of adjacent articular cartilage that may result in premature osteoarthritis”
Authors: Thank you, we have changed this.
Epidemiology
· The epidemiology in this rare disease is difficult to be specific and certain about. The increased risk in blacks is seen in a single study and likely biased due to the population surveyed. I would therefore qualify this statement.
· This is also true with the statement about basketball as the primary sport.
Authors: Thank you for this point. We have deleted statement about black ethnicity and revised sentence about sport.
· The picture and MRI of the hip ‘OCD’ is interesting. The possibility that this is a true OCD is low and the chondral shearing does not appear to be the type of pathology seen. Given that the hip is perhaps the most rare location for OCDs I would suggest this be eliminated from the manuscript to reduce confusion.
Authors: We have deleted Figure 1 as suggested.
Clinical Picture
· This section is well written
Authors: Thank you.
· Again, the example of a navicular OCD is confusing. Interesting but confusing and may not be an actual OCD, in my opinion. I would consider eliminating this example.
Authors: We have deleted Figure 4 as suggested.
· Classification systems – Classification systems are still evolving and are somewhat confusing when compared to one another. Using a classification system from 1959 is not, in my opinion, helpful except for historical purposes.
Authors: Thank you for this suggestion we have replaced Berndt&Harty classification by Lefort from 2006.
Treatment
· Non-operative treatment for stable lesions in young patients is often successful. The limited discussion of this should be expanded.
Authors: Thank you, we have expanded this section as suggested.
· The surgical treatment of OCDs is detailed. It is difficult to understand and perhaps could be combined into a table.
Authors: We appreciate your suggestion to shorten the treatment section. However, it's important to note that a previous reviewer had recommended extending this section, and we have invested considerable effort based on that feedback to enhance its comprehensiveness and detail. Consequently, we would like to confirm if shortening it is indeed necessary. Additionally, we have already removed Table 3 as suggested by another reviewer. We look forward to your guidance on how best to balance these differing viewpoints to optimize the manuscript.
Conclusion
· Agreed
Authors: Thank you.
Reviewer 2 Report
Comments and Suggestions for Authors While the authors present a thorough and up-to-date unstructured narrative review on OCD for the casually familiar audience, it should still contain a Methods section. This reviewer did not find any indication how this review was carried out. The authors might want to say "skyline or sunrise view". PRISMA guidelines have a version for this type of review, and those should be followed. Comments on the Quality of English Language While the prose is generally very readable and clear, proofreading is needed to address periodic grammar and typographical errors.Author Response
While the authors present a thorough and up-to-date unstructured narrative review on OCD for the casually familiar audience, it should still contain a Methods section. This reviewer did not find any indication how this review was carried out. The authors might want to say "skyline or sunrise view". PRISMA guidelines have a version for this type of review, and those should be followed.
Authors: Thank you for this suggestion we have added Methods section and provided important changes in title.
Reviewer 3 Report
Comments and Suggestions for Authors
Congratulations for the paper. My comments are:
· The abstract must have the sections that the journal guide proposes.
· The authors must write a more complete introduction to establish the justification of the work at the end. The authors must explain the importance of doing this review and what this paper contributes.
· Line 90: authors should change the word "precise" to another more appropriate word
· The authors should establish the design of the study/paper in the title and the objective of the paper.
· In section 4, authors must use a word other than "picture".
· Table 3 is not in correct format. The table must provide different information than the text. If the information is similar, I would delete it.
· Authors must include a clinical applications section in order to determine the practical purpose for therapists.
· The style of references is correct.
Comments on the Quality of English LanguageMinor editing of English language required
Author Response
Congratulations for the paper. My comments are:
· The abstract must have the sections that the journal guide proposes.
Authors: Thank you, we have added structures into abstract.
· The authors must write a more complete introduction to establish the justification of the work at the end. The authors must explain the importance of doing this review and what this paper contributes.
Authors: Thank you for this suggestion, we have added some explanation what our paper contributes.
· Line 90: authors should change the word "precise" to another more appropriate word
Authors: Authors: We have changed this.
· The authors should establish the design of the study/paper in the title and the objective of the paper.
Authors: Thank you for this suggestion, we have changed this.
· In section 4, authors must use a word other than "picture".
Authors: We have changed this.
· Table 3 is not in correct format. The table must provide different information than the text. If the information is similar, I would delete it.
Authors: We have removed Table 3 as suggested.
· Authors must include a clinical applications section in order to determine the practical purpose for therapists.
Authors: Thank you, we have incorporated such section as suggested.
· The style of references is correct
Reviewer 4 Report
Comments and Suggestions for Authors
The manuscript deals with an overview of the status and treatment of Osteochondritis Dissecans.
The authors have done a great synthesis of the literature on this topic; however, I miss some section on children-adults (apart from the table) and also the differences between men and women beyond epidemiology.
On the other hand, regarding conservative treatment, no treatment has been specified, including physiotherapy treatment, patient education itself and pharmacological treatment too. Could you please add some references on this?
Congratulations for the huge number of references used.
Author Response
The manuscript deals with an overview of the status and treatment of Osteochondritis Dissecans.
The authors have done a great synthesis of the literature on this topic; however, I miss some section on children-adults (apart from the table) and also the differences between men and women beyond epidemiology.
Authors: Thank you, we have added some information on the differences between juvenile and adult OCD. However, we were unable to find substantial literature on the differences between men and women beyond epidemiological data.
On the other hand, regarding conservative treatment, no treatment has been specified, including physiotherapy treatment, patient education itself and pharmacological treatment too. Could you please add some references on this?
Authors: Thank you for this suggestion, we have extended section dedicated to conservative treatment an provide some new references.
Congratulations for the huge number of references used.
Authors: Thank you.
Round 2
Reviewer 2 Report
Comments and Suggestions for Authors
This reviewer appreciates the efforts the authors have taken to revise the manuscript, especially adding a Methods section. Unfortunately, the Methods section is insufficient for a reader to be able to replicate the authors' search and article selection process which makes the Methods section insufficient. The authors are again advised to follow the PRISMA 2020 guidelines; although, this reviewer does understand that the analysis portions of the PRISMA guidelines are not relevant to this narrative review. Thank you for the opportunity to review this revised manuscript that provides a much-needed overview of OCD. I look forward to reading the authors' next revision.
Comments on the Quality of English LanguageAdditional careful proofreading is needed. The authors have continued to use direct quotations and continue to have inappropriate hyphenations. Direct quotes should be reworded in the authors' verbiage and the hyphenation should be removed.
Author Response
Thank you for taking the time to read our article and for appreciating our efforts. We have revised our methods section and, among other updates, have added inclusion and exclusion criteria. We followed the guidelines outlined in the following publications: doi: 10.1179/2047480615Z.000000000329doi:10.1016/S0899-3467(07)60142-6 doi: 10.4300/JGME-D-22-00480.1 We hope that with these revisions, the article is now suitable for publication.
Reviewer 3 Report
Comments and Suggestions for Authors
The authors have responded to all my comments correctly.
Author Response
Thank you very much for the advice.
Best regards
Reviewer 4 Report
Comments and Suggestions for Authors
Dear authors,
Thank you to following my suggestions.
Author Response

(The authors gave the same response as above.)
